# Metabolic oscillations on the circadian time scale in *Drosophila* cells lacking clock genes

Guillaume Rey[1,*,†,‡] iD, Nikolay B Milev[1,†,§], Utham K Valekunja[1], Ratnasekhar Ch[1,2], Sandipan Ray[1,2], Mariana Silva Dos Santos[1] iD, Andras D Nagy[1] iD, Robin Antrobus[3], James I MacRae[1] & Akhilesh B Reddy[1,**] iD

## Abstract

Circadian rhythms are cell-autonomous biological oscillations with a period of about 24 h. Current models propose that transcriptional feedback loops are the primary mechanism for the generation of circadian oscillations. Within this framework, *Drosophila* S2 cells are regarded as "non-rhythmic" cells, as they do not express several canonical circadian components. Using an unbiased multi-omics approach, we made the surprising discovery that *Drosophila* S2 cells do in fact display widespread daily rhythms. Transcriptomics and proteomics analyses revealed that hundreds of genes and their products, and in particular metabolic enzymes, are rhythmically expressed in a 24-h cycle. Metabolomics analyses extended these findings and demonstrate that central carbon metabolism and amino acid metabolism are core metabolic pathways driven by protein rhythms. We thus demonstrate that 24-h metabolic oscillations, coupled to gene and protein cycles, take place in nucleated cells without the contribution of any known circadian regulators. These results therefore suggest a reconsideration of existing models of the clockwork in *Drosophila* and other eukaryotic systems.

**Keywords** circadian; clocks; metabolomics; proteomics; transcriptomics
**Subject Categories** Genome-Scale & Integrative Biology; Metabolism; Quantitative Biology & Dynamical Systems
**Mol Syst Biol. (2018) 14: e8376**

## Introduction

In the fruit fly *Drosophila melanogaster*, the recognised models of the circadian clock centre on the transcription factors CYCLE (CYC) and CLOCK (CLK), the homologs of BMAL1 and CLOCK in mammals (Young & Kay, 2001). These control the transcription of several clock genes including *period* (*per*) and *timeless* (*tim*) (Panda *et al*, 2002). Only a subset of cells in *Drosophila* expresses these clock components and it is regarded as the principal pacemaker that drives daily activity rhythms and physiology. All other cells are not thought to have the capacity to generate 24-h rhythms autonomously. This extends to *Drosophila* Schneider 2 (S2) cells, which are one of the most commonly used fly cell lines, originally derived from a primary culture of late-stage embryos (Schneider, 1972). These cells are regarded as non-rhythmic because they do not express key circadian clock components, including PER, TIM or CLK, and therefore do not have the necessary apparatus to form a transcription–translation feedback loop to drive 24-h oscillations (Saez & Young, 1996; Darlington *et al*, 1998). Given that several lines of evidence indicate that circadian oscillatory behaviour is not fully dependent on clock genes (Lakin-Thomas, 2006), we set out to investigate whether such a "clock-less" system could exhibit 24-h oscillations. To this end, we adopted an unbiased multi-omics approach to comprehensively determine the daily dynamics of gene expression and metabolic state in this cellular model.

## Results

### Defining the daily transcriptome in *Drosophila* S2 cells

We first characterised gene expression patterns in S2 cells using RNA Sequencing (RNA-Seq). After synchronising cells with daily temperature cycles (Glaser & Stanewsky, 2007) for a week, we sampled cells at 3-h intervals in constant conditions (at 25°C in darkness) and measured their transcriptome by RNA-Seq (Appendix Fig S1A). We used a mixture model to define the set of expressed transcripts (Appendix Fig S1B), which established that several clock genes including *clk*, *per* and *tim* were not expressed in S2 cells (Fig 1A), consistent with previous studies

---

1   The Francis Crick Institute, London, UK
2   UCL Institute of Neurology, London, UK
3   Cambridge Institute for Medical Research (CIMR), Wellcome Trust/MRC Building, Addenbrooke's Hospital, Cambridge, UK
    *Corresponding author. Tel: +44 203 7963351; E-mail: guiomrey@gmail.com
    **Corresponding author. Tel: +44 203 7963426; E-mail: areddy@cantab.net
    †These authors contributed equally to this work
    ‡Present address: Department of Genetic Medicine and Development, University of Geneva Medical School, Geneva, Switzerland
    §Present address: Center for Integrative Genomics, Geńopode, University of Lausanne, Lausanne, Switzerland

---

(Saez & Young, 1996; Darlington *et al*, 1998). Moreover, four canonical clock genes that are expressed in S2 cells—*clockwork orange* (*cwo*), *cyc, par domain protein 1* (*pdp1*) and *vrille* (*vri*)— did not exhibit 24-h rhythmicity (Appendix Fig S1C). This verified that any known circadian components were either absent or not rhythmic in this cell line.

In contrast, we detected 482 rhythmic transcripts with a period of approximately 24 h in the same cells using the JTK-Cycle algorithm (Hughes *et al*, 2010; Fig 1B and C; adjusted *P*-value < 0.05), with peak phases at CT0 and CT12. Most transcripts had a relatively low amplitude of oscillation (Appendix Fig S1D), similarly to previous RNA-Seq studies in *Drosophila*, which showed that the majority of transcripts have an amplitude of two-fold or less (Hughes *et al*, 2012). We used a permutation-based method (Xie *et al*, 2005) to estimate the false discovery rate (FDR) and found that at least two-thirds of the rhythmic transcripts that we detected

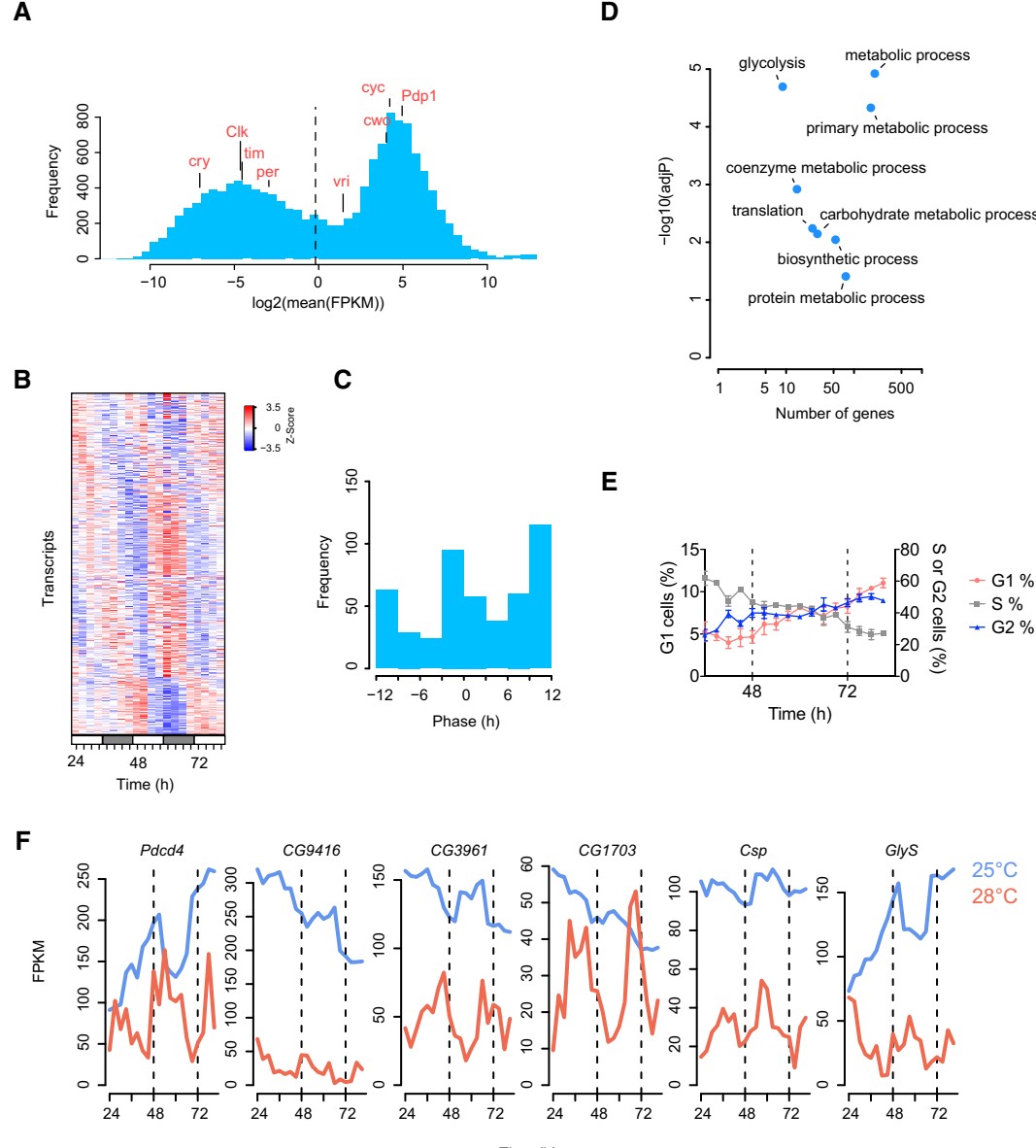

**Figure 1.  Transcriptional oscillations in *Drosophila* S2 cells.**

A   Histogram showing the distribution of mean gene expression levels. The dashed vertical line represents the cut-off chosen to define the set of expressed transcripts. FPKM, fragments per kilobase of transcript per million mapped reads.

B   Heatmap showing the expression profiles (ordered by phase) of 482 rhythmic transcripts (JTK-Cycle, *P* < 0.05, FDR = 0.31).

C   Phase distribution of circadian transcripts shown in (B).

D   Scatterplot representation of Gene Ontology (GO) analysis of the rhythmic transcripts.

E   Cell cycle analysis using flow cytometry showing the fraction of cells in G1, S and G2 phases over a time course experiment (*n* = 3 per time point, mean ± SEM).

F   Example of six transcripts detected as rhythmic under constant conditions at 25 or 28°C (JTK-Cycle, *P* < 0.05).

are likely to be truly rhythmically expressed (FDR = 0.31, $n$ = 10,000; Appendix Fig S1E and Materials and Methods). Importantly, we excluded the presence of low amplitude temperature fluctuations in the incubators used for culturing cells, which could have led to the appearance of artefactual daily oscillations (Appendix Fig S1F). Furthermore, we validated rhythmicity using two alternative detection methods [RAIN (Thaben & Westermark, 2014) and ARSER (Yang & Su, 2010)], which showed a highly significant overlap between the groups (Appendix Fig S1G; two-sided Fisher's exact test, $P < 10^{-16}$ for every pairwise comparison). In addition, we compared the set of 482 rhythmic transcripts with a meta-analysis of five circadian transcriptome studies in *Drosophila* (Keegan *et al*, 2007) and found 18 genes in common (Appendix Fig S1H). Thus, without any known clock genes being rhythmic, 24-h gene expression cycles were readily apparent in S2 cells.

Next, we performed Gene Ontology (GO) analyses to identify the functions of rhythmically expressed transcripts. Strikingly, rhythmic genes were enriched for protein biosynthesis and metabolic processes including glycolysis, a principal pathway in glucose metabolism (Fig 1D). To validate the rhythmic expression of metabolic transcripts, we measured the mRNA accumulation of three glycolytic genes—*lactate dehydrogenase* (*ImpL3*), *enolase* (*eno*) and *glyceraldehyde 3-phosphate dehydrogenase 2* (*Gapdh2*)—using quantitative PCR (qPCR). We found similar temporal profiles to our RNA-Seq data (Appendix Fig S1I). Of note, *ImpL3* was recently found to be expressed in a circadian pattern in *Drosophila* heads using identical statistical criteria to those used in our study, which substantiates our approach and could imply daily regulation of glycolysis in whole animals (Kuintzle *et al*, 2017).

Unlike most mammalian cell lines, S2 cells do not exhibit contact inhibition in culture and were actively dividing during our time courses. We therefore next determined whether the cell cycle contributed to the rhythmic transcription that we observed. We performed flow cytometry using DAPI staining to define the cell cycle status of temperature-synchronised cells in a 2-day time course. There was no 24-h variation in the proportion of cells in G1, S and G2 phases (Fig 1E and Appendix Fig S2A), indicating that the phasing of the circadian and cell cycles is not correlated. Moreover, we also measured the density of cells during the time course experiment and found a doubling time of 37 h, indicating that the cell cycle is not likely to contribute to daily transcriptional oscillations (Appendix Fig S2B). In order to validate these findings using an independent method, we used stably transfected S2 cells with the Fly-FUCCI system (Zielke *et al*, 2014). We measured the fraction of fixed cells in each phase of the cell cycle using flow cytometry and reached similar conclusions (Appendix Fig S2C and D). In addition, we also assembled the mRNA profiles of transcripts related to the cell division using GO annotations for mitotic cell cycle. Daily patterns of gene expression were not apparent in this set of transcripts, and only 10 cell cycle transcripts were statistically detected as rhythmic (JTK-Cycle, adjusted *P*-value < 0.05; Appendix Fig S2E and F). Together, this implied that the cell cycle does not contribute in any significant way to 24-h gene expression profiles.

An important property of circadian clocks is their "temperature compensation", whereby the period of the clock does not speed up or slow down significantly when maintained at constant high or low temperatures, respectively (Konopka *et al*, 1989). To determine whether S2 cell rhythms exhibited this phenomenon, we performed a low-coverage RNA-Seq time course experiment at 28°C (that is, at 3°C higher than the original time course). S2 cells did not tolerate higher temperatures than this, restricting the temperature range that we could use experimentally. We found 144 (7%) of the 2,035 transcripts measured displayed 24-h rhythmicity (JTK-Cycle, period 21–27 h, adjusted *P*-value < 0.05; FDR = 0.31; Appendix Fig S3A–C). If there was no temperature compensation, increased temperature should lead to shorter period of oscillations. To test this, we determined the number of rhythmic transcripts with periods between 12 and 18 h. In contrast to the 24-h analysis, we found only 71 transcripts in this shorter period range (JTK-Cycle, adjusted *P*-value < 0.05), and more importantly, the FDR was 0.99, implying that all were likely to be false positives (Appendix Fig S3D). This indicates that 24-h oscillations are dominant at 28°C. We performed the same analysis of rhythmicity for the 25°C time course and observed very similar results (periods between 12 and 18 h, 194 transcripts, adjusted *P*-value < 0.05, FDR = 1; Appendix Fig S3E). In addition, we determined the overlap between the sets of rhythmic transcripts detected at 25 and 28°C (Fig 1F and Appendix Fig S3F). Notably, overlap became significant as soon as the threshold of the JTK algorithm was increased above 0.1 (one-sided Fisher's exact test, Appendix Fig S3G) and the fraction of shared transcripts increased in proportion with the threshold of the JTK algorithm (Appendix Fig S3H). In a similar fashion to the 25°C experiments, we did not observe overt periodicity in cell cycle-related transcripts and found only six cell cycle transcripts rhythmically expressed (Appendix Fig S3I and J). Collectively, these results suggest that an uncharacterised mechanism, independent of canonical circadian genes or the cell cycle, is involved in the generation of temperature-compensated 24-h transcriptional oscillations in S2 cells.

## The daily proteome is enriched for abundant proteins with low amplitude oscillations

Gene Ontology (GO) had revealed that, in addition to metabolic processes, there was an overrepresentation of transcripts involved in protein translation and protein metabolic processes (Fig 1D). This led us to hypothesise that global changes in protein levels might be occurring on a 24-h timescale. To test this, we first quantified total protein content over a time course and found that it displayed a rhythmic pattern (Appendix Fig S4A). Given this, we proceeded to determine whether specific proteins were regulated rhythmically using multiplexed quantitative proteomics (Fig 2A). We quantified 4,759 proteins over 18 time points and determined that 342 (7%) of these were rhythmically expressed (Fig 2B and Appendix Fig S4B; JTK-Cycle, adjusted *P*-value < 0.05, FDR = 0.28). As before, we employed two further rhythm detection methods to the data and found a similar number of rhythmic proteins (Appendix Fig S4C; two-sided Fisher's exact test, $P < 10^{-16}$ for every pairwise comparison). As was the case for RNA transcripts, there was a biphasic distribution of phases (Fig 2C). However, most proteins peaked at CT0 (Fig 2C), while the peak phase for transcripts was CT12 (Fig 1B and C). Together these results indicate that S2 cells generate 24-h gene expression cycles in tandem with rhythmic regulation of the translation of specific proteins.

Next, we validated protein quantification using an alternative strategy for multiplex labelling (Appendix Fig S4D). We then correlated temporal profiles generated by our initial and alternative labellings, which demonstrated that both protocols yield reproducible results (Appendix Fig S4E and F), and the overlap between proteins in the quantification and validation sets was highly significant (Appendix Fig S4G; two-sided Fisher's exact test, $P < 5 \times 10^{-4}$). For example, ImpL3, pyruvate carboxylase (PCB) and fat-spondin, all displayed almost identical profiles using either quantification method (Fig 2D). Similarly to previous studies in mammals (Mauvoisin *et al*, 2014; Robles *et al*, 2014), the amplitude of protein oscillations was relatively low (Fig 2E). Interestingly, however, rhythmic proteins are more likely to be the more abundant ones quantified (Fig 2F). Notably, we again found that metabolic processes were functionally overrepresented among rhythmic proteins (Fig 2G), in a similar way to rhythmic transcripts (Fig 1D). Together, these results suggest that the rhythmic proteome in S2 cells is enriched for abundant metabolic enzymes and thus identify cellular metabolism as a key process regulated in a cyclical manner.

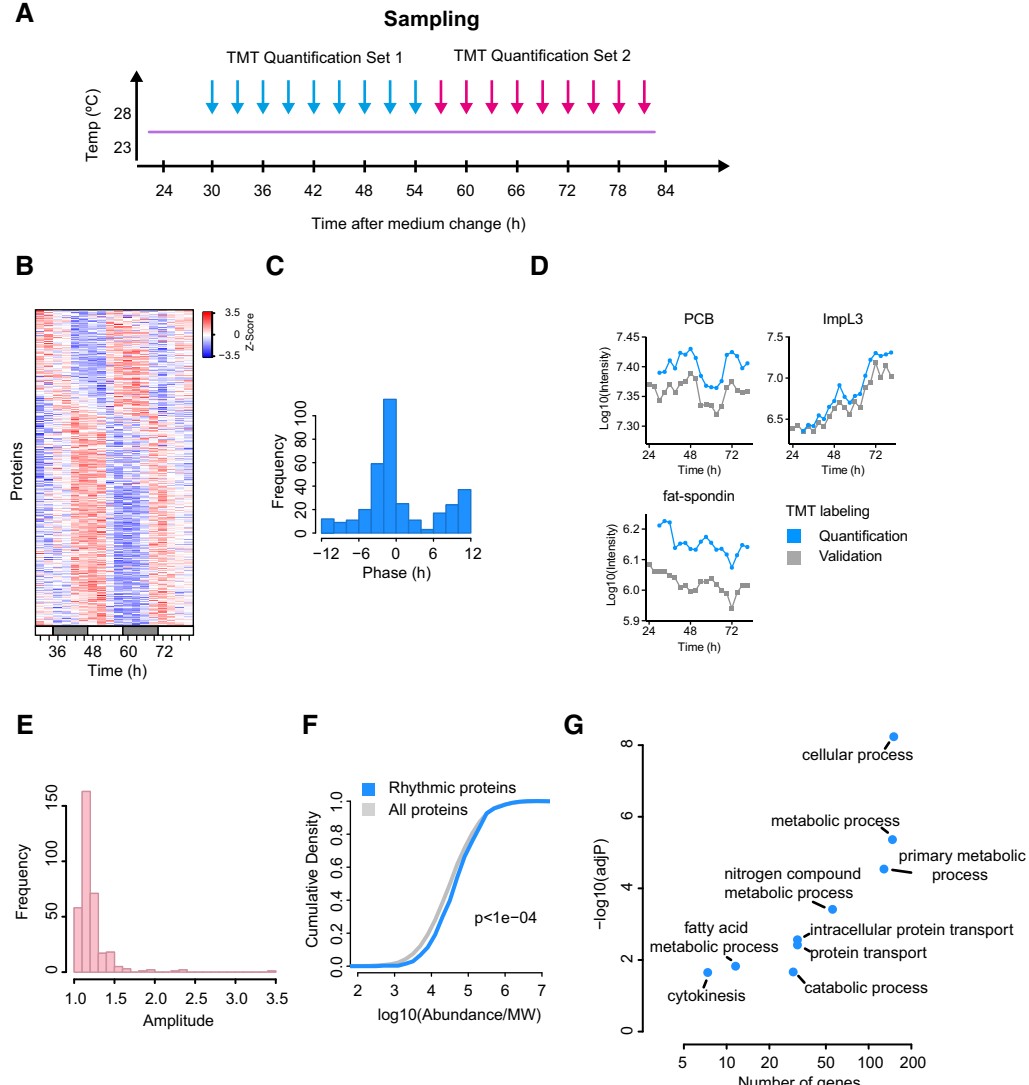

**Figure 2.  Proteome oscillations in *Drosophila* S2 cells.**

A    Sample collection procedure and labelling scheme using tandem mass tag (TMT) proteomics.

B    Heatmap representation of 342 rhythmic proteins (JTK-Cycle, $P < 0.05$).

C    Phase distribution of the circadian proteins shown in (B).

D    Validation of TMT quantitation using an alternative method to label samples.

E    Distribution of amplitudes of the 342 circadian proteins. Amplitudes were calculated by taking the ratio between the maximum and the minimum of each protein profile.

F    Cumulative density of protein abundances of circadian proteins vs. all proteins ($n$ = 4,759 proteins quantified; Wilcoxon rank-sum test, $P < 10^{-4}$).

G    Scatterplot representation of the GO analysis of the rhythmic proteins.

## Different contributions of transcriptional and post-transcriptional mechanisms to the daily proteome

Given that transcripts must be translated into proteins, the temporal profiles of mRNA and protein at a global scale might be expected to be highly coupled. To test this, we first correlated the abundance of transcripts and proteins, which revealed a tight correlation (Fig 3A). Despite this, there was little overlap of the sets of rhythmic transcripts and proteins (Fig 3B). To understand why this might be, we correlated the four categories of RNA-protein pairs (e.g. rhythmic RNA vs. rhythmic protein and so on). This showed that the correlation between transcript and protein is significantly higher when either the transcript or the protein is rhythmic, or if they both are (Fig 3C), which we validated independently by temporal cross-correlation (Appendix Fig S5A). Detrending of the RNA and protein profiles increased the correlation between RNA-protein pairs, in particular for pairs for which both profiles are rhythmic, further highlighting that 24-h

patterns contribute significantly to the correlation structure (Appendix Fig S5B).

We next examined the relationship of the key circadian properties, amplitude and phase, between RNA-protein pairs. Amplitudes were related in a similar way as before, with high correlation when both RNA and protein were rhythmic (Appendix Fig S5C). However, the phase of transcripts was a poor predictor of protein phase. If highly related, there should be a narrow distribution of phase differences (lags), which was not the case (Fig 3D and E). By contrast, rhythmic proteins were expressed in phase with their transcripts or with a phase delay of about 12 h (Fig 3F and G). This phase delay is likely related to the fact that most transcripts are expressed around CT12 (Fig 1C), while most proteins peak around CT0 (Fig 2C). To determine whether there were more complex underlying temporal relationships, we performed principal component analysis (PCA) on each data set. Interestingly, ordered temporal transitions between four different transcriptional states were mostly captured by the first two PCA components (Appendix Fig S5D). When considering

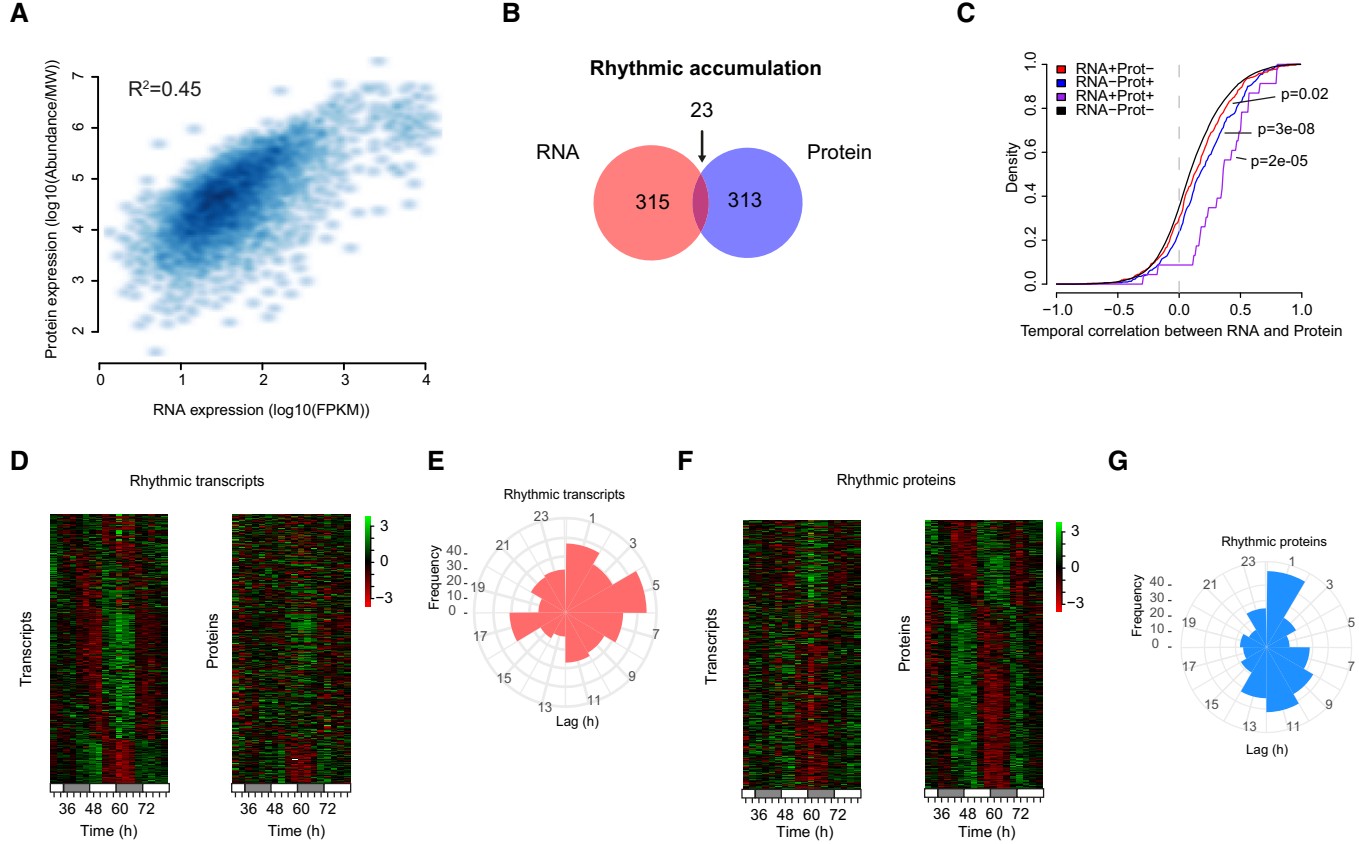

**Figure 3. Integration of transcriptomics and proteomics data.**

A  Scatter plot of RNA and protein expression.

B  Venn diagram showing the overlap of rhythmic RNA and proteins.

C  Cumulative distributions of Pearson's correlation coefficients for the indicated groups of RNA-protein pairs. RNA⁻Prot⁻, neither circadian; RNA⁺Prot⁺, both circadian; RNA⁻Prot⁺, RNA not circadian and protein circadian; RNA⁺Prot⁻, RNA circadian and protein not circadian (Wilcoxon rank-sum test).

D  Heatmap representations of transcripts and protein accumulations for the RNA-protein pairs with rhythmic transcripts.

E  Distribution of phase difference between RNA-protein pairs for those with rhythmic transcripts.

F  Heatmap representations of transcripts and protein accumulations for the RNA-protein pairs with rhythmic proteins.

G  Distribution of phase difference between RNA-protein pairs for those with rhythmic proteins.

matching protein profiles, we saw similar clustering of time points, but this time with three states showing a cyclic relationship (Appendix Fig S5E). Taken together, these results underline a complex combination of transcriptional and post-transcriptional mechanisms must regulate mRNA processing and degradation to shape the 24-h proteome, consistent with previous studies in mammalian systems (Reddy *et al*, 2006; Mauvoisin *et al*, 2014; Robles *et al*, 2014).

### Central carbon metabolism and amino acid metabolism exhibit daily oscillations

Given that rhythmic proteins were functionally enriched for metabolic processes (Fig 2G), we hypothesised that metabolic oscillations might be driven by 24-h rhythms in proteins. To test this, we did a similar time course to before and then performed untargeted liquid chromatography–mass spectrometry (LC-MS) to determine the ensemble of oscillating metabolites in S2 cells. We detected 1,339 features, among which 466 (35%) were rhythmic (Fig 4A and Appendix Fig S6A–C; JTK-Cycle, adjusted *P*-value < 0.05; FDR = 0.18). In keeping with our gene expression data, rhythmic features were clustered into two phases (Fig 4B and C), indicating a high degree of temporal organisation. To definitively identify a subset of metabolites, we next performed tandem MS (MS2), which yielded 54 metabolites (4% of those detected) with which we could perform pathway enrichment analysis (Fig 4D, and Appendix Fig S6D and Appendix Table S1; metabolite set enrichment analysis, raw *P*-value < 0.05; Xia & Wishart, 2010). Significantly, central carbon metabolism and pathways associated with amino acid metabolism were enriched among rhythmic metabolites. We validated regulation of central carbon metabolism by performing targeted LC-MS analysis of glycolysis, pentose phosphate pathway and citric acid cycle metabolites. Of note, key metabolites including ATP, glutathione and citric acid cycle intermediates exhibited rhythmic accumulation over time (Fig 4E and F, and Appendix Table S2).

To determine mechanistic relationships between rhythmic protein expression and metabolic oscillations, we correlated the two. We first considered all possible protein–metabolite associations, which resulted in a non-uniform distribution with an overrepresentation of highly correlated and anti-correlated profiles (Appendix Fig S6E). When only the best match between each metabolite and associated proteins was kept, the effect was much more pronounced, since most protein–metabolite pairs had an absolute correlation coefficient greater than 0.5 (Appendix Fig S6F). For example, lactate dehydrogenase (*ImpL3*) and NAD, its substrate,

displayed strongly correlated profiles (Fig 4G; Pearson's correlation = 0.97). In contrast, phosphoribosylamidotransferase (Prat), an essential enzyme in the pathway for *de novo* purine synthesis, was anti-correlated to the levels of its product glutamate (Pearson's correlation = −0.82). Using a similar strategy, we analysed the correlation between proteins and targeted metabolites in central carbon metabolism (Fig 4H). This demonstrated that most protein–metabolite pairs were correlated, especially those found in the same subpathway. For example, proteins and metabolites of the pentose phosphate pathway formed a small cluster of correlated profiles, as exemplified by sedoheptulose 7-phosphate (SH7P) and its cognate enzyme, transaldolase (Taldo; Appendix Fig S6G). Collectively, these results suggest a robust link between protein rhythmicity and downstream 24-h metabolite oscillations in fundamental biochemical pathways.

## Discussion

We have shown that even though *Drosophila* S2 cells do not express the core components of transcriptional feedback loops thought to be required for rhythm generation in the fly, they nevertheless exhibit temperature-compensated 24-h metabolic oscillations coupled to gene and protein cycles. Our results thus indicate that the canonical circadian network may not be required to generate genome-wide oscillations on the circadian time scale in a eukaryotic system (Appendix Fig S7A). The fact that CLK and CYC are transcription factors with a PAS domain, which allows molecular sensing of the intracellular environment, is consistent with a role as an important connecting cog between metabolic and gene expression programmes. In this interpretation, *Drosophila* S2 cells are thus a novel model of cellular time keeping that encompasses metabolic and gene expression components, but not the canonical circadian clock gene network. We speculate that cells and tissue in the fly that do not express clocks genes, such as *per* and *tim*, may nonetheless exhibit similar daily rhythms. This is highly relevant since *Drosophila* have been thought to be somewhat unique among model organisms, from single-celled cyanobacteria to mammals, in not having a clock in every cell. Our results suggest that *Drosophila* may in fact have a clock in every cell, but not necessarily based on the presence of, or oscillation of, clock genes such as *per* or *tim*. Understanding the relative role of gene, protein and metabolite expression in this model will enhance our understanding of fundamental properties of circadian oscillations in *Drosophila* and in other eukaryotic systems.

---

**Figure 4.  The circadian metabolome of S2 cells.**

A   Heatmap representation of the 466 LC-MS features detected as rhythmic (JTK-Cycle; *P* < 0.05, FDR = 18%).
B   Individual traces of rhythmic features from cluster 1 and cluster 2 from (A).
C   Distribution of phases from (A).
D   Metabolite set enrichment analysis of the 54 identified rhythmic metabolites.
E   Targeted LC-MS analysis of metabolites from glycolysis, pentose phosphate pathway (PPP) and tricarboxylic acid (TCA) cycle. The colour of each node represents the *P*-value for daily rhythmicity (JTK-Cycle). See Appendix Table S2 for list of abbreviations.
F   Selected temporal profiles of metabolites are shown together with their associated *P*-value.
G   Two examples of protein–metabolite pairs are shown with their respective Pearson's correlation coefficients.
H   Heatmap showing the Pearson's correlation between each profile from targeted metabolomics (carbon metabolism) and proteomics data.

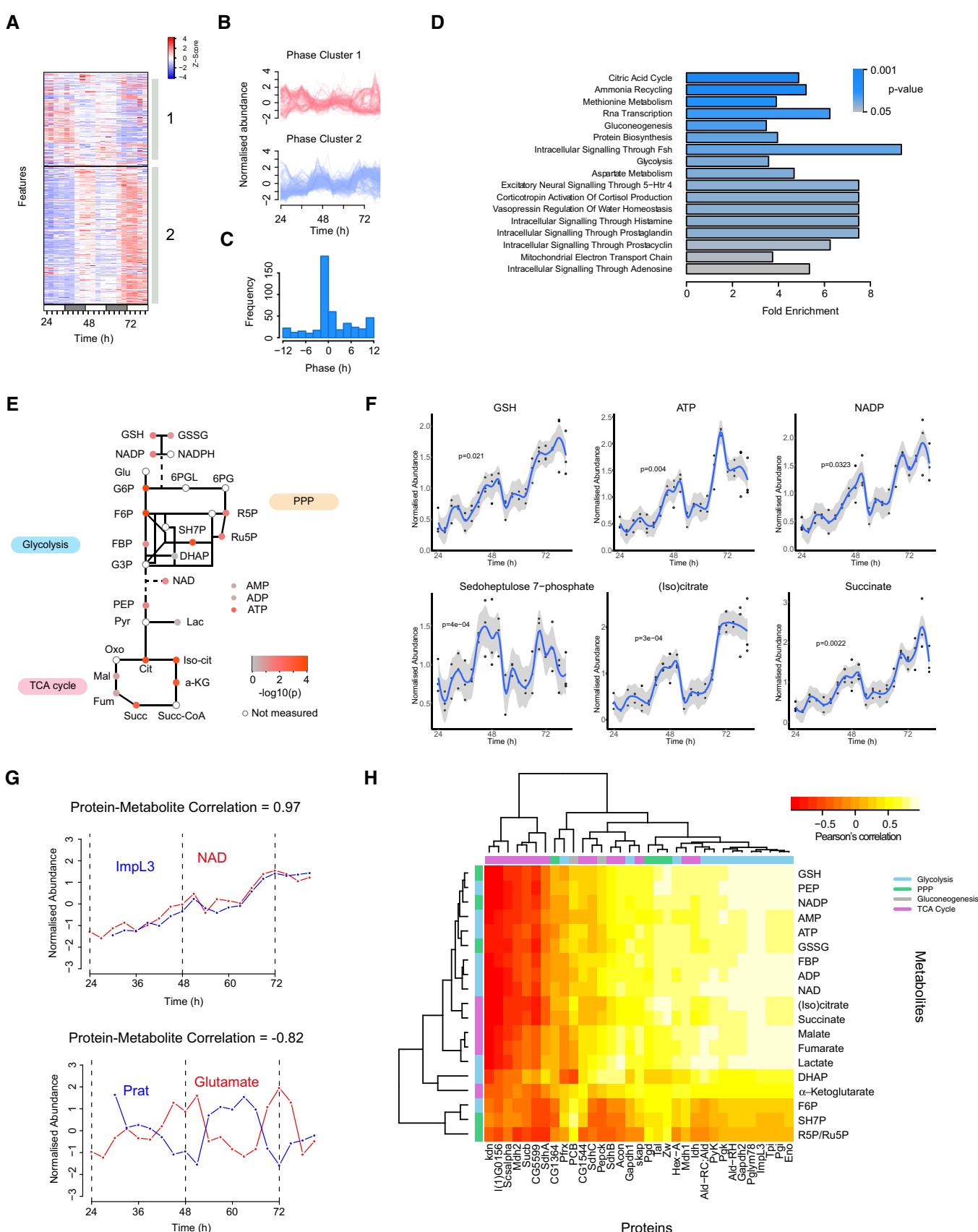

**Figure 4.**

# Materials and Methods

### Cell culture

*Drosophila* S2 cells were purchased from Thermo Fisher Scientific and were grown at 25°C in Schneider's *Drosophila* Media (Thermo Fisher Scientific), supplemented with 10% heat-inactivated FBS, 1% Pen-Strep and 1/500 MycoZap Plus-CL. For circadian entrainment protocol, S2 cells were subjected to temperature cycles (12 h at 23°C, 12 h at 28°C) for at least 1 week, with media changes occurring every 3–4 days at the transition between 23 and 28°C. The last medium change was performed at $t = 0$ h, and cells were plated into six-well plates. Cells were kept at 25°C for the remaining of the experiment, with sample collection occurring at 3-h intervals between 24 and 81 h. For the RNA-Seq experiment at 28°C, S2 cells were grown at 28°C in Schneider's *Drosophila* Media, supplemented with 10% heat-inactivated FBS and 1% Pen-Strep. Cells were plated into six-well plates and were synchronised with 2 days of temperature cycles (12 h at 28°C, 12 h at 23°C) followed by 24 h at 28°C, before the start of sample collection at $t = 24$ h. To measure the change in cell density over the time course, we used the Countess Automated Cell Counter (Invitrogen) system following manufacturer's instructions.

### RNA isolation and RNA sequencing

At the time points indicated in the main text, cells were lysed in triplicate in TRI-Reagent, flash-frozen and stored at −80°C until extraction. Extraction and purification were performed with the Direct-zol RNA MiniPrep kit (Zymo Research). RNA-Seq libraries for the 25 and 28°C time courses were respectively prepared using polyA selection method (KAPA Stranded mRNA-Seq Kit for Illumina Platforms, Roche) or previously published rRNA-depletion method (Rey *et al*, 2016). Sequencing using a HiSeq platform with single-end 50-bp reads and subsequent quality filtering of reads was performed according to manufacturer's instructions (Illumina).

### RNA-seq data analysis

Sequencing reads were aligned to the UCSC *Drosophila* reference genome (dm3) using TopHat v2.1.0 (Kim *et al*, 2013). For the RNA-Seq experiment at 25°C, we obtained on average 30.0 million mapped reads per sample, with a minimum of 16.2 millions. Reads were assembled into transcripts using RefSeq genes as reference and their abundance estimated using Cufflinks/Cuffmerge/Cuffquant/Cuffnorm v2.2.1 (Trapnell *et al*, 2010). To estimate the threshold to define the set of expressed transcripts, we modelled the distribution of transcript mRNA expression using a Gaussian mixture model using the R package "mixtools". We chose a threshold of 0.9 FPKM, which corresponds to the 0.95 percentile of the distribution of lowly expressed transcripts, to define a set of 6,944 expressed transcripts. For the RNA-Seq experiment at 28°C, aligned reads were subsampled to have a maximum of 1 million alignments per sample (with a minimum of 0.5 million), in order to compensate for differences in sequencing depth between barcodes. To define the set of expressed transcripts in the 28°C RNA-Seq data, we used the same threshold of 0.9 FPKM, as the 28°C time course had a lower coverage that did not allow to estimate accurately the distribution of lowly expressed transcripts.

Temporal profiles were linearly detrended by fitting of a straight line to each profile and subtracting the resulting function. The JTK-Cycle algorithm was used to detect rhythmic transcripts using the following parameters: minimal period = 21, maximal period = 27 and adjusted *P*-value = 0.05. As validation, two alternative algorithms, RAIN (period = 24, period.delta = 3, method = longitudinal, and *P*-value = 0.01) and ARSER (minimal period = 21, maximal period = 27, default period = 24 and *P*-value = 0.01), were used to detect rhythmic transcripts. To determine the FDR of identification of rhythmic transcripts, we used a permutation-based method that removes positives to reduce bias in the estimation of the FDR (Xie *et al*, 2005). These permutation tests were run 10,000 times, and FDR was estimated by taking the ratio between the mean number of rhythmic profiles in the permuted samples and the number of rhythmic profiles in the original ordering. An empirical *P*-value was also computed based on the distribution of rhythmic profiles in the permutated samples. GO analyses were performed using the Gene List Analysis Tool from the PANTHER database using all *D. melanogaster* genes as reference set (Mi *et al*, 2016). GO annotation of transcripts for mitotic cell cycle (GO:0000278) was retrieved from FlyBase.org and was used to visualise transcript profiles of cell cycle-related genes.

### Quantitative PCR

Triplicate RNA samples were reverse-transcribed into cDNA using the High Capacity cDNA Reverse Transcription Kit (Life Technologies), following the manufacturer's instructions, using 0.3–1 μg total RNA per reaction. The resulting cDNAs were used in duplicate 7 μl PCR, set up as follows: 3.5 μl TaqMan Gene Expression Master Mix (Life Technologies), 0.35 μl validated TaqMan Gene Expression Assay (Life Technologies), 1.15 μl nuclease-free water and 2 μl cDNA. Real-time PCR was performed with an ABI 7900HT (Applied Biosystems) system. The following TaqMan gene expression assays were used: Eno (Dm01844953), Gapdh2 (Dm01843776), ImpL3 (Dm01841229) and Act5C (Dm02361909). The relative levels of each mRNA were calculated by the $2^{-\Delta C_t}$ method and normalised to the corresponding Act5C levels.

### Flow cytometry

At the time points mentioned in the main text, cells were washed twice in PBS and resuspended in ice-cold 70% ethanol. Cells were kept at 4°C until staining. For staining DNA, cells were first washed twice with PBS, resuspended in DAPI solution (1 μg/ml in PBS + 0.1% Triton) and kept overnight at 4°C. For quantitation of cell cycle phase transitions using fluorescent reporters, S2 cells were transfected with a *Drosophila*-specific system (Fly-FUCCI), a commercially available multicistronic vector for expression of fluorescent ubiquitin-based cell cycle indicators (Addgene 73164; Zielke *et al*, 2014). The Fly-FUCCI system allows G1, S and G2 phases of interphase in cultured *Drosophila* cells to be distinguished by relying on fluorochrome-tagged degrons from the Cyclin B (RFP-tagged) and E2F1 proteins (GFP-tagged). Freshly split cells were transfected in six-well plates using 2 μg of DNA and 6 μg of linear 25 kDa Polyethylenimine (Alfa Aesar). Fluorescent probes expressed in S2 cells were visually checked under a fluorescent microscope (EVOS FL Cell Imaging System, Thermo Fisher Scientific). Selection of expressing cells was

done using Geneticin (Thermo Fisher Scientific) for 2 weeks. At the time points mentioned in the main text, cells were washed twice in PBS and resuspended in 4% formaldehyde and stored at 4°C until analysis. Cell cycle phase distribution was assessed by flow cytometry, where GFP single-positive cells showed G1, RFP single positives indicated S phase and double positives represented G2 phase populations. Flow cytometry was performed on a LSRFortessa™ Cell Analyzer (BD Biosciences) using standard methods.

### Proteomics sample preparation

At the time points indicated in the main text, cells were spun down and the pellets were flash-frozen and stored at −80°C until protein extraction. To extract protein, pellets were lysed on ice with 500 μl of Lysis Buffer [100 mM triethylammonium bicarbonate (TEAB), 1% SDS, 1% NP-40, 10 mM diethylenetriaminepentaacetic acid (DTPA), 1/100 Halt protease inhibitors (Thermo Fisher Scientific)]. Cells were vortexed and incubated for 30 min on ice. Samples were sonicated using a Bioruptor Standard (Diagenode) for 5 min (30 s On, 30 s Off) on medium power. Samples were spun at max speed at 4°C for 10 min to remove debris and transferred to fresh tubes. BCA assay (Thermo Fisher Scientific) was used to quantify protein levels for tandem mass tag (TMT) labelling (Thermo Fisher Scientific).

Tandem mass tag labelling was performed according to manufacturer's instructions. 200 μg per condition was transferred into a new tube, and the volume was adjusted to 200 μl with 100 mM TEAB. 10 μl of 200 mM TCEP was added to each sample to reduce cysteine residues, and samples were incubated at 55°C for 1 h. To alkylate cysteines, 10 μl of 375 mM iodoacetamide was added to each sample and samples were incubated for 30 min protected from light at room temperature. Samples were split in two, and acetone precipitation was performed by adding six volumes (~600 μl) of pre-chilled (−20°C) acetone. The precipitation was allowed to proceed overnight at −20°C. The samples were centrifuged at 8,000 × g for 10 min at 4°C, before decanting the acetone.

Acetone-precipitated (or lyophilised) protein pellets were resuspended with 100 μl of 100 mM TEAB. 2.5 μg of trypsin per 100 μg of protein was added to the proteins for proteolytic digestion. Samples were incubated overnight at 37°C to complete the digestion. TMT Label Reagents were resuspended in anhydrous acetonitrile, and 0.4 mg of each label was added to the corresponding peptide sample. The reaction was allowed to proceed for 1 h at room temperature. 8 μl of 5% hydroxylamine was added to each sample and incubated for 15 min to quench the labelling reaction. Samples were combined in a new microcentrifuge tube at equal amounts and stored at −80°C until mass spectrometry analyses.

### Proteomics mass spectrometry

Tandem mass tag-labelled tryptic peptides were subjected to HpRP-HPLC fractionation using a Dionex UltiMate 3000 powered by an ICS-3000 SP pump with an Agilent ZORBAX Extend-C18 column (4.6 × 250 mm, 5 μm particle size). Mobile phases ($H_2O$, 0.1% $NH_4OH$ or MeCN, 0.1% $NH_4OH$) were adjusted to pH 10.5 with the addition of formic acid, and peptides were resolved using a linear 40 min 0.1–40% MeCN gradient over 40 min at a 400 μl/min flow rate and a column temperature of 15°C. Eluting peptides were collected in 15 s fractions. One hundred and twenty fractions

covering the peptide-rich region were re-combined to give 12 samples for analysis. To preserve orthogonality, fractions were combined across the gradient. Re-combined fractions were dried down using an Eppendorf Concentrator (Eppendorf, UK) and resuspended in 15 μl MS solvent (3% MeCN, 0.1% TFA).

Data for TMT-labelled samples were generated using an Orbitrap Fusion Tribrid Lumos mass spectrometer (Thermo Scientific). Peptides were fractionated using an RSLCnano 3000 (Thermo Scientific) with solvent A comprising 0.1% formic acid and solvent B comprising 80% MeCN, 20% $H_2O$ and 0.1% formic acid. Peptides were loaded onto a 75 cm Acclaim PepMap C18 column (Thermo Scientific) and eluted using a gradient rising from 7 to 37% solvent B by 180 min at a flow rate of 250 nl/min. MS data were acquired in the Orbitrap at 120,000 fwhm between 380 and 1,500 m/z. Spectra were acquired in profile with AGC $2 \times 10^5$. Ions with a charge state between $2^+$ and $7^+$ were isolated for fragmentation using the quadrupole with a 0.7 m/z isolation window. CID fragmentation was performed at 35% collision energy with fragments detected in the ion trap between 400 and 1,200 m/z. AGC was set to $1 \times 10^4$, and MS2 spectra were acquired in centroid mode. TMT reporter ions were isolated for quantitation in MS3 using synchronous precursor selection. Ten fragment ions were selected for MS3 using HCD at 65% collision energy. Fragments were scanned in the Orbitrap at 60,000 fwhm between 120 and 500 m/z with AGC set to $1 \times 10^5$. MS3 spectra were acquired in profile mode.

### Proteomics data analysis

MaxQuant v1.5.5.1 (Cox & Mann, 2008) was used to process the raw TMT proteomics data using the following parameters: fixed modifications = carbamidomethylation, FDR for protein and peptide identification = 0.01, sequence database = UniprotKB proteome for *D. melanogaster* (downloaded on 13 January 2017), variable modifications = oxidation of methionine, protein N-terminal acetylation. TMT 10plex data were normalised using linear regression in logarithmic space to normalise for global abundance variations between each TMT channel. For the quantification TMT sets, samples were TMT-labelled as shown in Fig 2A (set1: pooled sample, CT30, CT33, CT36, CT39, CT42, CT45, CT48, CT51 and CT54; set2: pooled sample, CT57, CT60, CT63, CT66, CT69, CT72, CT75, CT78 and CT81). For each protein, the quantification TMT set1 and set2 were assembled together after taking the log ratio between each channel and the reference channel (pooled sample of all time points). For the validation TMT sets, samples were TMT-labelled as shown in Appendix Fig S2C (set1: CT24, CT30, CT36, CT42, CT48, CT54, CT60, CT66, CT72 and CT78; set2: CT27, CT33, CT39, CT45, CT51, CT57, CT63, CT69, CT75 and CT81). For each protein, the validation TMT set1 and set2 were assembled together after taking the log ratio between each channel and the mean across all channels for this protein (mean centring). Protein temporal profiles were linearly detrended by fitting of a straight line to each profile and subtracting the resulting function. The JTK-Cycle algorithm (Hughes *et al*, 2010) together with the RAIN and ARSER methods for validation was used to detect rhythmic transcripts with same parameters used for the RNA-Seq analysis. To determine the FDR of identification of rhythmic proteins, we used the same method as for transcripts. GO analyses were performed using the Gene List Analysis Tool from the PANTHER database using all *D. melanogaster* genes as reference set

(Mi *et al*, 2016). In order to integrate transcriptomics and proteomics data, we used the UniProt ID converter tools to generate a mapping between UniProt accession numbers and RefSeq annotations. We performed a protein-centric integration, where each protein was associated with at most one transcript. If there were more than one transcript, the most rhythmic mRNA transcript was kept. Using this strategy, we were able to assemble 4,658 protein-RNA pairs from 4,758 proteins and 6,944 transcripts. For each RNA-protein pair, the Pearson correlation coefficient was computed using the respective temporal profiles from CT30 to CT81.

## Metabolite sample preparation

At the time points indicated in the main text, triplicate cell samples were spun down and pellets were washed with room temperature PBS. Cells were resuspended in 1 ml of methanol:water (80:20) at $-75°C$ for quenching, and samples were stored at $-80°C$. To extract metabolites, samples were thawed on ice and then vortexed for 2 min at room temperature. Samples were sonicated for 5 min in the cold room (30 s ON, 30 s OFF, medium power). Samples were centrifuged for 10 min at 9,391 *g* at 4°C, and the supernatants were transferred to fresh 1.5-ml Eppendorf tubes. The extraction was repeated a second time with 500 μl methanol:water (80:20). For the third extraction, 500 μl methanol:water (80:20) supplemented with $^{13}C_5,^{15}N_1$-valine was used downstream quality control. The three extractions were pooled and lyophilised to dryness. Samples were resuspended in 350 μl of chloroform:methanol:water (1:3:3 v/v), and the polar (upper) phase was collected for analysis. Quality control samples were prepared by pooling equal volumes from all samples included in this study.

## Metabolomics mass spectrometry

LC-MS method was adapted from a published protocol (Zhang *et al*, 2012). Samples were injected onto a Dionex UltiMate LC system (Thermo Scientific) with a ZIC-pHILIC (150 × 4.6 mm, 5 μm particle) column (Merck SeQuant). A 15-min elution gradient of 80–20% Solvent B was used, followed by a 5 min wash of 5% Solvent B and 5-min re-equilibration, where Solvent B was acetonitrile (Optima HPLC grade, Sigma-Aldrich) and Solvent A was 20 mM ammonium carbonate in water (Optima HPLC grade, Sigma-Aldrich). Other parameters were as follows: flow rate 300 μl/min; column temperature 25°C; injection volume 10 μl; and autosampler temperature 4°C. MS was performed with positive/negative polarity switching using an Q Exactive Orbitrap (Thermo Scientific) with a HESI II probe. MS parameters were as follows: spray voltage 3.5 and 3.2 kV for positive and negative modes, respectively; probe temperature 320°C; sheath and auxiliary gases were 30 and 5 arbitrary units, respectively; and full scan range: 70–1,050 m/*z* with settings of AGC target and resolution as balanced and high (3 × 10⁶ and 70,000), respectively. Data were recorded using Xcalibur 3.0.63 software (Thermo Scientific). Mass calibration was performed for both ESI polarities before analysis using the standard Thermo Scientific Calmix solution. To enhance calibration stability, lock-mass correction was also applied to each analytical run using ubiquitous low-mass contaminants. Parallel reaction monitoring (PRM) acquisition parameters were the following: resolution 17,500; collision energies were set individually in HCD (high-energy collisional dissociation) mode.

## Metabolomics data analysis

Qualitative analyses were performed using Xcalibur Qual Browser (Thermo Fisher Scientific) and mzCloud (HighChem). Untargeted metabolomics data analyses were performed with Progenesis QI (Nonlinear Dynamics) using the following parameters: feature detection = high resolution and peak processing = centroided data with resolution at 70,000 (FWHM). In positive mode, the following adducts were used: M+NH₄, M+H, M+Na and M+2H. In negative mode, the following adducts were used: M-H, M+Na-H and M-2H. Normalisation was performed using the log-ratio method over all features. Features having a coefficient of variation (CV) lower than 30% among quality control samples were selected for downstream analyses (*n* = 722 and 616 for positive and negative mode, respectively). PCA of all samples (including features with CV < 30% from positive and negative modes) shows very good clustering, indicating system stability, performance and reproducibility (Appendix Fig S4A). Similar conclusions were reached using correlation analysis (Appendix Fig S4B). Features in the retention time window between 19.15 and 19.35 min were excluded from subsequent analyses, due to artefactual profiles in this time window. Temporal profiles were linearly detrended by fitting of a straight line to each profile and subtracting the resulting function. The JTK-Cycle algorithm was used to detect circadian rhythmicity using the following parameters: minimal period = 21, maximal period = 27, adjusted *P*-value = 0.05 and number of replicates = 2–3. To determine the FDR of identification of rhythmic proteins, we used the same method as for transcripts. From the 466 rhythmic features, 145 with at least one hit in spectral databases were selected for MS2 annotation. Out of these, we were able to annotate 70 features with MS2 data (Appendix Table S1), which correspond to 54 metabolites. Metabolic pathway enrichment analysis was performed using metabolite set enrichment analysis (MSEA; Xia & Wishart, 2010). For targeted LC-MS data analysis, a set of 20 metabolites (Appendix Table S2) was chosen from carbon metabolism and redox pathways. Retention time and MS/MS spectra from samples were compared to metabolite standards to validate identification. Quantification was performed manually using TraceFinder v4.1 (Thermo Fisher Scientific). Normalisation across samples was performed using the normalisation ratio calculated with Progenesis QI. In order to integrate metabolomics and proteomics data sets, we used the Kyoto Encyclopedia of Genes and Genomes (KEGG) annotation. Briefly, UniProt accession numbers were annotated with Enzyme Commission (EC) numbers, which were used to fetch all interacting metabolites in the KEGG database. Each metabolite was annotated with all possible proteins based on the described annotation, and correlation analysis was performed between metabolite–protein pairs.

## Data availability

The RNA-seq data sets produced in this study has been deposited in the Gene Expression Omnibus (accession number GSE102495). The mass spectrometry proteomics data have been deposited to the ProteomeXchange Consortium via the PRIDE (Vizcaíno *et al*, 2016) partner repository with the dataset identifier PXD007669.

**Expanded View** for this article is available online.

## Acknowledgements

ABR is supported by The Francis Crick Institute, which receives its core funding from Cancer Research UK (FC001534), the UK Medical Research Council (FC001534) and the Wellcome Trust (FC001534 and 100333/Z/12/Z). ABR also acknowledges funding from the European Research Council (ERC Starting Grant No. 281348, MetaCLOCK), the EMBO Young Investigators Programme and the Lister Institute of Preventive Medicine. GR was supported by an Advanced SNSF Postdoctoral Mobility Fellowship and an EMBO Long-Term Fellowship.

## Author contributions

GR, NBM and ABR designed and planned the experiments. GR, NBM, UKV and RC performed cell experiments. GR, NBM and UKV performed the RNA-Seq experiments and GR analysed the data. GR, NBM and ADN performed the flow cytometry experiments. GR, NBM and SR performed the proteomics experiments and GR analysed the data. RA performed the mass spectrometry analyses for proteomics samples. GR and RC performed the metabolomics extractions. MSDS performed the mass spectrometry analyses for metabolomics samples, under the supervision of JIM. GR performed the metabolomics data analyses, under the supervision of MSDS and JIM. GR and ABR wrote the manuscript, with contributions from all of the authors.

## Conflict of interest

The authors declare that they have no conflict of interest.

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
