## [Review Process File · Molecular Systems Biology]

Metabolic oscillations on the circadian time scale in *Drosophila* cells lacking clock genes

Guillaume Rey, Nikolay B. Milev, Utham K. Valekunja, Ratnasekhar Ch, Sandipan Ray, Mariana Silva Dos Santos, Andras D. Nagy, Robin Antrobus, James I. MacRae, and Akhilesh B. Reddy.

Review timeline:

Submission date:	11 th April 2018
Editorial Decision:	7 th May 2018
Revision received:	12 th June 2018
Accepted:	16 th July 2018

Editor: Maria Polychronidou.

Transaction Report:

1st Editorial Decision

7th May 2018

Thank you again for submitting your work to Molecular Systems Biology. We have now heard back from the two referees who agreed to evaluate your study. As you will see below, the reviewers think that the presented findings are likely to be of interest for the field. They raise however a series of concerns, which we would ask you to address in a revision.

One of the more fundamental points raised refers to the need to perform additional analyses in order to exclude that the observed oscillations are driven by the growth conditions or cell cycle effects. All other issues raised need to be convincingly addressed. Please let me know in case you would like to discuss further any of the comments of the reviewers.

REFeree REPORTS.

Reviewer #1:

The authors present a fascinating study of rhythmic gene expression, protein abundance, and metabolism in cultures of S2 cells, usually thought to have no circadian rhythm. This is an important and timely study because of the emerging realization that some rhythmicity can persist in eukaryotic cells in the absence of the classical TTFL based on canonical clock genes.

This study consists of impressive transcriptomic, proteomic, and metabolomic data sets collected over 60 hours, providing a much clearer molecular picture than studies with more limited sampling windows. The statistical analyses are generally well done.

Given the amount of effort and care that went into the -omics portion of the study, I wish that more attention had been paid to the growth of the cultures over the course of the experiment. As the

authors are clearly aware, an obvious alternative explanation, instead of a cryptic circadian oscillator, is some combination of the cell cycle and non-constant growth conditions. The authors do show the fraction of cells in various cell cycle stages using DAPI staining + flow cytometry in Fig. 1E, but the interpretation of these data are not clear, since the cell cycle status of the culture is clearly changing during the experiment. It seems that growth is slowing down on a timescale of ~2 days.

Though it seems unreasonable to request additional -omics data sets, the authors should do more to verify that the rhythms they observe are not generated by growth / cell cycle effects, perhaps by probing selected transcripts. Ruling this out seems central to their interpretation. Does another oscillation occur beyond 84 hours? Can oscillations be detected if the culture medium is not changed at $t=0$? Are the oscillations phase shifted if the 25 C and 28 C entraining protocol is inverted? Perhaps the culture medium could be refreshed more frequently, to achieve a semi-chemostatic condition. Certainly, the authors should report the doubling time in their hands at both 25 C and 28 C and the change in density of cells over the course of the experiment.

A more minor point: I found the discussion of the rhythm analysis in 25 C vs 28 C conditions confusing. The authors argue that JTK-cycle gives no strong statistical signal for rhythms between 15-21 hours in 28 C conditions (but they don't give the equivalent analysis for 25 C?) It would be much more informative to try to estimate the typical period of oscillation from the transcriptomic data (w/ error bar) in both 25 C and 28 C conditions and compare these to each other and to the culture doubling times.

Reviewer #2:

This intriguing paper reports a comprehensive 'omics profiling of temperature-entrained, *Drosophila* S2 cell cultures after transfer to constant temperature. ~7000 'expressed' RNAs, 4759 Proteins and 1399 metabolite species are tested for rhythmicity, using 20 RNA-seq timepoints in triplicate; 18 proteomics timepoints in two experiments; and 20 metabolomics timepoints. I take the aim of the paper to be the detection of rhythmicity in these cell cultures, rather than a comprehensive survey. That aim is significantly advanced. The rhythms are low in amplitude but this is expected. Hence, the credibility of the analysis method is important: several were used in each case, and they are the best available, and they agree for a good proportion of RNAs and proteins. The permutation tests for FDR are critical, an evaluation of that approach in these cases will be important and I don't provide it here.

This remains an early-stage report but with significant potential interest to the circadian field, given the increasing interest in non-canonical oscillator mechanisms. The *Drosophila* system has rarely been tested for such rhythms at the molecular level. Some secondary claims here are not well supported but they could be modified without great loss to the significance of the manuscript.

Major issues.

1. As the experimental designs are expected to detect a fraction of the true-rhythmic molecular species (and for the metabolites and proteins, the detected fraction of species is also relatively small), the overlaps between independent experiments can be low. Though greater overlap among 'omic-scale studies would be reassuring, it is not essential for the aim of the paper. However, it also means that the result will be hard to replicate, as a replicate would not necessarily overlap strongly with the present results. Double the number of data points would strongly enhance the power in rhythm detection (Hogenesch lab results) to avoid that problem. However, this is not very likely to be achieved; and in any case would be most valuable if an independent laboratory did the replication.

2. Sup Fig 1f does not eliminate the real possibility of covert, external driving signals reaching the cells and causing the rhythms observed. That would require a functional test where at least two intentional entrainment phases are shown to determine subsequent phase in free-run. This is a standard procedure and the authors should have done it; any rhythmic marker could be tested. However, this study would ideally be conducted during the key data-generating experiments. There is less value in performing it now, to show that temperature cycles CAN set phase in this

experimental setup, not that they DID so during the sampling for RNA, protein or other rhythms.

The same study would also demonstrate entrainability, which is a "canonical" circadian property. This goal is distinct from the avoidance of artefacts in the free-running data. For these two reasons, this single additional experiment would add most to the manuscript.

3. The higher-temperature analysis is limited by the short and sparsely-sampled data (both in time resolution and read depth). It is a useful addition but tests few of the RNAs that were rhythmic at 25C. It tests periods less than 21h, after a 3C temperature increase and finds less evidence for periods shorter than 21h compared to evidence for periods 21-27h. To calibrate our confidence in the result, it would be useful to know the equivalent comparison for shorter periods in the 25C data. However, I think the point is moot. The test demonstrates ongoing rhythmicity in the circadian range. It does not establish temperature compensation of any particular rhythm, which would require a more precise period estimate than is possible in this experimental design. Overall, the evidence for temperature compensation is too weak to support the current claims.

4. The paper cannot identify the "principal" rhythmic metabolic pathways in the cell (and perhaps not even in the data set) as only 54 metabolites were identified in MS2 (claim in the abstract).

Minor points

5. Supp 3a protein concentration in cell extracts needs further explanation. It would be very surprising if protein content per cell doubled every day. Protein concentration per volume of cell culture could increase as cells grew and divided. Growth might surprisingly but conceivably be rhythmic even if division wasn't.

6. GO analysis should not use the whole *Drosophila* genome as the background set, particularly for the proteomic analysis where only subset of proteins were quantified. Using rhythmic, quantified proteins as foreground compared to all possible proteins confounds enrichment by rhythmicity with enrichment by detection (some metabolic enzymes are abundant, this is not the question of interest). Rather, all the quantified proteins (or RNAs) should be the background set, removing the bias due to detectability and focussing instead on enrichment due to rhythmicity alone.

7. Introducing low and high expression (Supp 1b) is confusing if you subsequently refer to these gene sets as expressed/not expressed. The latter terms might be reconsidered, because low-expression spans a 1024-fold range of read counts, which is not obviously equivalent to no expression. The numbers in each category should be cited in text, so the proportion of rhythmic transcripts is clear.

8. Non-cyclic PCA clustering of the RNA data S4c deserves some comment. It might reflect a progressive change during the experiment, such that the second cycle of data did not replicate the first (see comments above on protein concentration), but there might also be other, specific explanations.

9. This raises one potential issue for the correlation analyses used in several figure panels: it would be important to correlate detrended data so that the correlation focuses on rhythmicity rather than the non-rhythmic trend, or to explain why this was not done.

10. Page 6, are there known causal connections between the correlated protein-metabolite pairs noted here? They are not at all obvious.

Typographical and stylistic issues.

11. "the overlap between the sets of rhythmic transcripts detected at 25{degree sign}C and 28{degree sign}C (Fig. 1g" there is no 1g.

12. It is very reassuring to have the two proteomic labelling strategies with a fairly high correlation Fig S3e. However, S3g's Venn diagram needs to include the numbers of quantified proteins that overlap between the two experiments as well as the rhythmic proteins.

13. Fig S5a, the clustering of the metabolomic QC samples is clear but the figure reveals nothing about the experimental samples unless the timepoints are labelled. The correlations in S5b don't summarise this as concisely as the PCA.

14. To show clustering of pathway proteins and metabolites in Fig 4h, the pathways should be labelled along the axes of the figure.

1st Revision - authors' response

12th June 2018

Point-by-Point Responses to Referees' Comments

Rey et. al "Metabolic oscillations on the circadian time scale in Drosophila cells lacking clock genes"

Molecular Systems Biology Manuscript MSB-18-8376

Reviewer #1:

The authors present a fascinating study of rhythmic gene expression, protein abundance, and metabolism in cultures of S2 cells, usually thought to have no circadian rhythm. This is an important and timely study because of the emerging realization that some rhythmicity can persist in eukaryotic cells in the absence of the classical TTFL based on canonical clock genes.

This study consists of impressive transcriptomic, proteomic, and metabolomic data sets collected over 60 hours, providing a much clearer molecular picture than studies with more limited sampling windows. The statistical analyses are generally well done.

Given the amount of effort and care that went into the -omics portion of the study, I wish that more attention had been paid to the growth of the cultures over the course of the experiment. As the authors are clearly aware, an obvious alternative explanation, instead of a cryptic circadian oscillator, is some combination of the cell cycle and non-constant growth conditions. The authors do show the fraction of cells in various cell cycle stages using DAPI staining + flow cytometry in Fig. 1E, but the interpretation of these data are not clear, since the cell cycle status of the culture is clearly changing during the experiment. It seems that growth is slowing down on a timescale of ~2 days.

Though it seems unreasonable to request additional -omics data sets, the authors should do more to verify that the rhythms they observe are not generated by growth / cell cycle effects, perhaps by probing selected transcripts. Ruling this out seems central to their interpretation.

We thank the reviewer for these comments. We measured the cell density over the time course at 25C and found a linear increase in cell density, thereby indicating a stable growth rate during the experiment (Appendix Figure S2B). This therefore indicates that the reported oscillations are not associated with non-constant growth conditions. We also found that the doubling time at 25C was much larger than 24h (about 37h), supporting the hypothesis that the cell division cycle does not contribute to daily gene expression oscillations in this system in any significant way.

Furthermore, we also performed an additional experiment using the FUCCI system to measure the fraction of cells in the different phases of the cell cycle during our time course (Appendix Figure S2C and D). This method therefore represents an independent experimental assessment of the cell cycle, by an independent method. Similarly to the DAPI data for cell cycle phases, which was in the initial submission, we did not observe a daily change in the respective fractions using this alternative method.

Moreover, we have assembled the profiles of cell cycle related transcripts in the two transcriptomics time courses using the Gene Ontology annotation for "mitotic cell cycle" and did not observe 24 h oscillations in this gene set (Appendix Figure S2E and F and Appendix Figure S3I and J).

Together, these further data rule out any detectable contribution of the cell cycle to the generation of daily oscillations in S2 cells.

Does another oscillation occur beyond 84 hours? Can oscillations be detected if the culture medium is not changed at $t=0$? Are the oscillations phase shifted if the 25 C and 28 C entraining protocol is inverted? Perhaps the culture medium could be refreshed more frequently, to achieve a semi-chemostatic condition. Certainly, the authors should report the doubling time in their hands at both 25 C and 28 C and the change in density of cells over the course of the experiment.

We thank the reviewer for thinking about these types of perturbations, which we think would be highly interesting to follow-up in future work. However, we feel that performing a differential experiment for temperature entrainment would involve generating another two sets of transcriptomics data which, as the reviewer themselves suggests, is unreasonable (given the costs and time this would take to complete).

We believe that trying to change the medium more frequently would likely reset the oscillations, as this is an important synchronization mechanism in eukaryotic cells, but we do not feel that analyzing this would add anything to the main findings of the paper (but represents a valuable area for future work).

We found that the oscillations are robust to changes in the growth protocol, as the time courses at 25 and 28C differ in their growth and entrainment protocol. In the 28C protocol, CT00 corresponds to the last change in temperature, with the last medium change occurring 48 hours before CT00. In contrast, in the 25C protocol, the last medium change occurs at CT00, together with the last change of temperature. This shows that the oscillations can occur after 84 hours and also if the medium change occurs before temperature entrainment.

A more minor point: I found the discussion of the rhythm analysis in 25 C vs 28 C conditions confusing. The authors argue that JTK-cycle gives no strong statistical signal for rhythms between 15-21 hours in 28 C conditions (but they don't give the equivalent analysis for 25 C?) It would be much more informative to try to estimate the typical period of oscillation from the transcriptomic data (w/ error bar) in both 25 C and 28 C conditions and compare these to each other and to the culture doubling times.

We apologize for the confusion in our discussion around this point. We have now included an analysis of oscillations of non-overlapping sets of periods (21-27 and 12-18 hours) for both time courses (Appendix Figure S1E and S3C-E). At both temperatures, we find a significant number of rhythmic transcripts around 24 h, while for short periods the false discover rate (FDR) is close to 1 (i.e. likely to be noise).

Reviewer #2:

This intriguing paper reports a comprehensive 'omics profiling of temperature-entrained, *Drosophila* S2 cell cultures after transfer to constant temperature. ~7000 'expressed' RNAs, 4759 Proteins and 1399 metabolite species are tested for rhythmicity, using 20 RNA-seq timepoints in triplicate; 18 proteomics timepoints in two experiments; and 20 metabolomics timepoints. I take the aim of the paper to be the detection of rhythmicity in these cell cultures, rather than a comprehensive survey. That aim is significantly advanced. The rhythms are low in amplitude but this is expected. Hence, the credibility of the analysis method is important: several were used in each case, and they are the best available, and they agree for a good proportion of RNAs and proteins. The permutation tests for FDR are critical, an evaluation of that approach in these cases will be important and I don't provide it here.

This remains an early-stage report but with significant potential interest to the circadian field, given the increasing interest in non-canonical oscillator mechanisms. The *Drosophila* system has rarely been tested for such rhythms at the molecular level. Some secondary claims here are not well supported but they could be modified without great loss to the significance of the manuscript.

Major issues.

1. As the experimental designs are expected to detect a fraction of the true-rhythmic molecular species (and for the metabolites and proteins, the detected fraction of species is also relatively small), the overlaps between independent experiments can be low. Though greater overlap among 'omic-scale studies would be reassuring, it is not essential for the aim of the paper. However, it also means that the result will be hard to replicate, as a replicate would not necessarily overlap strongly with the present results. Double the number of data points would strongly enhance the power in rhythm detection (Hogenesch lab results) to avoid that problem. However, this is not very likely to be achieved; and in any case would be most valuable if an independent laboratory did the replication.

We thank the reviewer for these comments. It has been noted previously that the overlap between transcriptomics studies in mammals and *Drosophila* is relatively low, most likely to insufficient statistical power to reliably detect rhythmic transcripts across experiments (i.e. there is a significant fraction of false negatives/positives).

In the case of the datasets presented in this paper (transcriptomics, proteomics and metabolomics), there is a high degree of convergence on molecular functions seen in rhythmic species. Moreover, this gives reassurance that the main conclusion of the paper – that there are non-canonical circadian rhythms in S2 cells – is sufficiently backed up by multiple different lines of experimental evidence and by different techniques.

2. Sup Fig 1f does not eliminate the real possibility of covert, external driving signals reaching the cells and causing the rhythms observed. That would require a functional test where at least two intentional entrainment phases are shown to determine subsequent phase in free-run. This is a standard procedure and the authors should have done it; any rhythmic marker could be tested. However, this study would ideally be conducted during the key data-generating experiments. There is less value in performing it now, to show that temperature cycles CAN set phase in this experimental setup, not that they DID so during the sampling for RNA, protein or other rhythms.

The same study would also demonstrate entrainability, which is a "canonical" circadian property. This goal is distinct from the avoidance of artefacts in the free-running data. For these two reasons, this single additional experiment would add most to the manuscript.

We thank the referee for these comments. As stated above, performing a differential experiment for temperature entrainment would involve generating another two sets of transcriptomics data, which is not reasonable as it will not change the major findings of the study.

Moreover, the experiments presented here have been very carefully designed to exclude external effects including light, temperature, acoustic and vibration cycles. Cells were placed in a temperature-controlled incubator in a room with constant temperature and constant light,

excluding potential feedback between incubator temperature control and environment. Doors of the incubator were only opened every 3 hours to collect plates for sampling. We would be interested to understand what other extraneous factors in a laboratory setting would need to be controlled for, that could account for an external driving signal in the 24 hours range.

3. The higher-temperature analysis is limited by the short and sparsely-sampled data (both in time resolution and read depth). It is a useful addition but tests few of the RNAs that were rhythmic at 25C. It tests periods less than 21h, after a 3C temperature increase and finds less evidence for periods shorter than 21h compared to evidence for periods 21-27h. To calibrate our confidence in the result, it would be useful to know the equivalent comparison for shorter periods in the 25C data. However, I think the point is moot. The test demonstrates ongoing rhythmicity in the circadian range. It does not establish temperature compensation of any particular rhythm, which would require a more precise period estimate than is possible in this experimental design. Overall, the evidence for temperature compensation is too weak to support the current claims.

We thank the reviewer for these comments. We have now included a similar analysis of oscillations at 12-18 hours for the 25C time course (Appendix Figure S3E). Even if our data are not able to precisely assess the period of oscillations at different temperatures, the expected changes for a cycle that is not temperature-compensated are large enough to be detected. If these rhythms were not temperature compensated, we would expect that the ~ 24-hour oscillation found at 25C should decrease to 17-20 hours at 28C for a Q10 in the range of 2-3 (which is typical for a normal biochemical process). However, we do not observe this change. This evidence therefore supports the contention that the oscillations are generated by a temperature-compensated cycle.

4. The paper cannot identify the "principal" rhythmic metabolic pathways in the cell (and perhaps not even in the data set) as only 54 metabolites were identified in MS2 (claim in the abstract).

We thank the reviewer for these comments. We have amended the text accordingly.

Minor points

5. Supp 3a protein concentration in cell extracts needs further explanation. It would be very surprising if protein content per cell doubled every day. Protein concentration per volume of cell culture could increase as cells grew and divided. Growth might surprisingly but conceivably be rhythmic even if division wasn't.

We thank the reviewer for these comments. The cell density is increasing over time (Appendix Figure S2B) and therefore the increase in total protein concentration reflects the increased cell density.

6. GO analysis should not use the whole Drosophila genome as the background set, particularly for the proteomic analysis where only subset of proteins were quantified. Using rhythmic, quantified proteins as foreground compared to all possible proteins confounds enrichment by rhythmicity with enrichment by detection (some metabolic enzymes are abundant, this is not the question of interest). Rather, all the quantified proteins (or RNAs) should be the background set, removing the bias due to detectability and focussing instead on enrichment due to rhythmicity alone.

We thank the reviewer for this suggestion. However, we do not agree that this is the best way to analyze the present data. A recent study has shown that highly expressed proteins are more likely to be rhythmic due to energetic constraints (Wang et al., Cell Reports 2015), and we observed an enrichment of abundant proteins among rhythmic ones. Given this, correcting for the selection of abundant proteins would also erroneously correct for rhythmic ones. To demonstrate this, when we did what is suggested, and used the set of detected proteins as background, we did not find any biological process significantly enriched (FDR<0.05).

7. Introducing low and high expression (Supp 1b) is confusing if you subsequently refer to these gene sets as expressed/not expressed. The latter terms might be reconsidered, because low-expression spans a 1024-fold range of read counts, which is not obviously equivalent to no expression. The numbers in each category should be cited in text, so the proportion of rhythmic transcripts is clear.

We apologize for not making this clear. We have changed the text to remove confusion between low and high expressed transcripts and those selected for further analysis.

8. Non-cyclic PCA clustering of the RNA data S4c deserves some comment. It might reflect a progressive change during the experiment, such that the second cycle of data did not replicate the first (see comments above on protein concentration), but there might also be other, specific explanations.

We thank the reviewer for thinking about these issues. The progressive change during the experiment found in the PCA analysis likely reflects non-circadian changes that take place during this 2.5 day time course, including a change in the metabolic environment in the (spent) medium and growth conditions.

9. This raises one potential issue for the correlation analyses used in several figure panels: it would be important to correlate detrended data so that the correlation focuses on rhythmicity rather than the non-rhythmic trend, or to explain why this was not done.

We thank the reviewer for these suggestions. We have now included an analysis of correlation of detrended RNA and protein profiles (Appendix Figure S5B). Removal of the linear trend improved the correlation of profiles, indicating that non-linear variation (including the 24h component) plays an important role. Stratifying profiles according to 24h rhythmicity allows us to look specifically at the effect of daily variation.

10. Page 6, are there known causal connections between the correlated protein-metabolite pairs noted here? They are not at all obvious.

We have changed the text accordingly to make these connections clearer.

Typographical and stylistic issues.

11. "the overlap between the sets of rhythmic transcripts detected at 25 {degree sign}C and 28 {degree sign}C (Fig. 1g" there is no 1g.

We thank the reviewer for pointing out this error. We have changed the text accordingly.

12. It is very reassuring to have the two proteomic labelling strategies with a fairly high correlation Fig S3e. However, S3g's Venn diagram needs to include the numbers of quantified proteins that overlap between the two experiments as well as the rhythmic proteins.

Using MaxQuant software, we analyzed the two datasets and therefore we report a single list of quantified proteins in the two datasets.

13. Fig S5a, the clustering of the metabolomic QC samples is clear but the figure reveals nothing about the experimental samples unless the timepoints are labelled. The correlations in S5b don't summarise this as concisely as the PCA.

Appendix Figure S6A was made to show the clustering of QC samples and their relative positions compared to all other samples. The correlations in Appendix Figure S6B show more precisely the correlations between all the samples.

14. To show clustering of pathway proteins and metabolites in Fig 4h, the pathways should be labelled along the axes of the figure.

We have changed the figure accordingly.

Corresponding Author Name: Akhilesh B. Reddy

Manuscript Number: MSB-18-8376